# The Frequent Use of Emergency Departments Among the Pediatric Population: A Retrospective Analysis in Rome, Italy

**DOI:** 10.3390/epidemiologia6030031

**Published:** 2025-06-21

**Authors:** Giuseppe Furia, Fabio Ingravalle, Antonio Vinci, Paolo Papini, Andrea Barbara, Patrizia Chierchini, Gianfranco Damiani, Massimo Maurici, Corrado De Vito

**Affiliations:** 1Department of Public Health and Infectious Diseases, Sapienza University of Rome, 00185 Rome, Italy; 2Local Health Authority Roma 1, Borgo Santo Spirito, 00193 Rome, Italy; 3Doctoral School in Nursing Sciences and Public Health, University of Rome “Tor Vergata”, 00133 Rome, Italy; 4Department of Life Sciences and Public Health, Università Cattolica del Sacro Cuore, Largo Francesco Vito 1, 00168 Rome, Italy; 5Department of Woman and Child Health and Public Health, Fondazione Policlinico Universitario “A. Gemelli” IRCCS, Largo Agostino Gemelli 8, 00168 Rome, Italy; 6Department of Biomedicine and Prevention, University of Rome “Tor Vergata”, 00133 Rome, Italy

**Keywords:** frequent user, children, appropriateness, local health authority, emergency medicine, population study, epidemiology

## Abstract

**Background/Objectives**: Inappropriate use of emergency department (ED) services is widely acknowledged to have a negative impact on health systems as a whole. A minor portion of Frequent User (FU) patients are often responsible for the disproportionate use of ED services. **Methods**: A retrospective population study was conducted on the attendances of pediatric EDs from 1 January 2022 to 31 December 2022 at the Roma 1 Local Health Authority in Rome, a territory served by more than 13 EDs. Nested logistic regression analysis was used to investigate patient characteristics in predicting inappropriate use of EDs. **Results**: In 2022, 35,691 pediatric ED attendances were recorded, with 24,824 patients distributed among 904 PCP/GPs in the six districts. A total of 71.8% of patients had only one attendance in 2022. A total of 3.5% of the patients were FUs, who were responsible for more than 10% of the attendances. However, most of the attendances were not appropriate. FUs, younger age (<2 yo), and more severe clinical presentations were more likely to be associated with appropriate ED attendance. PCPs/GPs and districts do not have a role in determining a higher number of ED attendances. The single patient’s characteristics have a greater influence on this phenomenon. **Conclusions**: Frequent use of the ED is not associated with inappropriate use among children, mainly due to the characteristics and needs of specific patients. However, parents’ education for a more rational use of health system resources and the provision of local solutions to children’s health needs may allow for a more appropriate use of health service resources.

## 1. Introduction

### 1.1. Background

Health systems must ensure equitable, integrated, and accessible healthcare services that meet user expectations. The emergency department (ED) represents the second level of care for administering immediate medical or surgical care to critically ill and acute patients. However, recent studies have reported the increasing utilization of the ED for non-urgent patients, both among adults and children < 18 years. The mean proportion of non-urgent pediatric ED attendances is 41.1% ± 15.2%, with children aged 4 years or under accounting for approximately 40% of all pediatric ED attendances and the highest rate registered by children under the age of 1 [1,2,3].

Available studies conducted in Italy report that non-urgent attendances account for 27.6% of ED users and 58.2% of total pediatric attendance episodes [4,5]. These percentages are greater than those of adults with inappropriate ED attendances (approximately 12%) [1,2,6]. Studies of overcrowding in the pediatric ED have identified important effects on quality of care, including delays in timely use of pain assessment scores for injuries, antibiotic administration for neonatal fever, and timely treatment of asthma [7,8,9]. Non-urgent attendances have been negatively associated with crowding and costs, causing longer waiting times and greater dissatisfaction among both parents and health workers [4,10].

There is no standard criterion for defining non-urgent ED attendances. The choice of threshold values is often subjective, but can be defined as attendances in a given time interval that do not require the specialized services of an ED that can be managed in other settings [1,11]. Furthermore, a quota of patients attends the ED more often than the general population and is defined as Frequent Users (FUs) [6]. The causes of frequent ED use are multifactorial. Demographic and social factors (e.g., age < 5, female gender, low income, health insurance, foreign nationality) and enabling factors such as the availability and affordability of healthcare services, parental attitudes and perceptions of the severity of illness, limited access to primary care, and proximity to the facility are all associated with non-urgent attendance [12].

Visiting the ED for non-urgent attendances can result in ED overcrowding, physician work overload, increased costs, and excessive use of investigations, increased risks of infections, and increased length of stay. This, coupled with the unavailability of beds in both secondary and tertiary care facilities, can lead to increased costs and dissatisfaction with healthcare services [13]. Frequent attendance behavior could be influenced by the emergency care models in each country, as well as the general healthcare system model and cultural differences between countries [14]. Strategies suggested to reduce non-urgent attendance include financial incentives such as higher co-payments, patient education, and redirection of patients to an alternative source of care and the provision of after-hours and weekend care by primary care physicians. Despite these efforts, non-urgent attendances have not decreased [1,2,15].

### 1.2. The Child Healthcare System in Italy

In Italy, the pediatric healthcare system is part of the National Health Service, with each region managing its own healthcare services depending on the needs of the specific population. In the Lazio region, where this study was conducted, healthcare is organized into three main levels of intervention: first access/primary care, secondary care/hospital care, and tertiary care based on specialty hospital care. Within primary care services, every Italian resident is registered with a Primary Care Pediatrician (PCP) or a General Practitioner (GP). Primary care includes general first-access care for children and adolescents (0–14 years), which is mandatory and provided by PCPs from 0 to 6, while parents can choose between a PCP and a GP for children who are between 6 and 14 years of age. From 15 years of age, primary care is provided by GPs [16,17].

The Local Health Authority (LHA) Roma 1 is one of the three LHAs in the urban area of Rome; it is one of the most populous areas in Italy. It contains 13 EDs, of which 8 are pediatric EDs, and approximately 170,000 residents were aged < 18 in 2022, with an aging index (number of populations aged > 64 years per 100 individuals aged < 14 years) of 196.0 (the Italian mean is 187.6) [18]. Its territorial articulation consists of six Local Health Districts (LHDs), each inhabited by 150,000–180,000 residents. LHDs guarantee primary care services to all residents, provided by the work of approximately 150 PCPs and 900 GPs.

The LHAs are part of the Italian National Health Service (NHS), which is publicly funded and run under a Beveridge model. Hospital emergency care is generally free of charge, although there may be some minimal fees if the patient does not actually need urgent care.

Many international studies have focused on the potential clinical causes of non-urgent attendances and the strategies and interventions available to reduce ED use, but few population studies have been conducted on children [2,19]. The primary objective of this study is to investigate in the ED the characteristics of pediatric attendances in Rome, to describe the characteristics of the pediatric population of FU, and to highlight the differences between FUs and non-FUs. The secondary objective is to identify the factors associated with appropriate attendance in the ED by comparing the level of variability attributed to the territorial component (LHD) and to the PCP/GP.

## 2. Materials and Methods

### 2.1. Study Design, Population, and Data Sources

A retrospective population study was conducted in 2024 to investigate the attendance of pediatric ED from 1 January 2022 to 31 December 2022. The total population aged < 19 yo (17.2% of total population, with 12.4% being <15 yo) was potentially eligible for inclusion. All patients aged 0–18 years who were residents of the LHA Roma 1 geographical area and who had at least 1 ED attendance in one of the 13 EDs were included. Patients who attended single-specialty emergency departments (ophthalmology and obstetrics) were excluded because they represent a population with specific needs and peculiarities worthy of specific focus. The 13 EDs comprise the total ED structures within the LHA Roma 1 territory. Data were extracted from the Healthcare Emergency Information System and collected in pseudo-anonymized form through a digital business intelligence platform routinely used by LHA Roma 1. Each individual’s ID was encrypted, allowing for the linkage of health events to the same person while preserving anonymity. Using these pseudo-anonymized identifiers, repeated attendances in 2022 were tracked, and patients were classified as either “FU” or “non-FU” based on their total number of visits.

The resulting records included the following information for each patient:Number of attendances: According to the definition of an adult FU, a pediatric FU is defined as having ≥4 attendances per year, according to the criteria used in previous studies and reviews of the literature [2,20,21].Demographic characteristics: The gender was classified as binary (male or female) according to the LHA registry; age was classified according to the age stages of the National Institute of Child Health and Human Development: “infancy” (birth to 12 months); “toddler” (13 to 24 months); “early childhood” (25 months to 5 years); “middle childhood” (6 to 12 years); and “adolescent” (12 to 15 years) [22].Arrival mode: Emergency Medical Services (EMS) or not EMS;Triage code: Color codes (1, 2, 3, 4, 5—from the most urgent to the least) [23];Festivity or workday: During weekends and holidays, primary care services are not available;Appropriateness of attendance: According to 2019 Italian Ministry of Health’s priority code triage system, ED attendances are considered “inappropriate” when it meets one of the following criteria: triage code “white” (noncritical, nonurgent patients) or “green” (low urgency and priority) and outcome “home discharge” or “leave during medical examinations” or “leave without being seen by the physician” [5,23];Patient’s PCP/GP (only doctors associated with at least one patient’s access to EDs were selected);Patient’s LHD.

All variables were mandatory in each patient’s attendance record, so no data were missing. The REporting of Studies Conducted using Observational Routinely Collected Health Data (RECORD) guidelines were used for study reporting [24].

### 2.2. Statistical Analysis

Statistical analysis was performed using Microsoft^®^ Excel^®^ v.2016 MSO and STATA v. 17.0. The cumulative number of ED attendance was calculated for each patient’s ID, and individuals with more than four attendances were classified as FUs. Descriptive analysis was performed on all recorded variables. A descriptive analysis was conducted on all recorded variables, using statistics such as mean, standard deviation (SD), frequency, and percentage to summarize the sample’s demographic and ED attendance characteristics. Time series analysis was conducted to identify weekly components, with particular interest for Saturdays and Sundays, which are not working days for GPs and PCPs. Since only one year of data was available, it was not possible to correct for other seasonality effects.

Given the large sample size, statistical significance was determined at a level of *p* = 0.001 for inferential analysis. Welch’s *t* test was used to compare the mean ages between FUs and non-FUs. A Chi-square test was used to investigate differences in categorical variables between FUs and non-FUs. A simple univariate logistic analysis was performed first to identify the factors potentially associated with adequate attendance in the ED. Then, a multivariate nested logistic regression model was built that included all the identified predictors. To understand whether the appropriateness of ED use was more influenced by organizational factors (PCP/GP or LHD) or individual factors, a four-level model was proposed, with observations clustered as follows:attendance < patient < PCP/GP < LHD

The role of individual variables was expressed as Odds Ratios (ORs); the role of grouping variables was expressed as Median Odds Ratio (MOR), calculated by Merlo et al., which summarizes the variability of ORs across different clusters, providing insight into the influence of group-level factors on the outcome variable [25,26].

## 3. Results

In 2022, 35,691 pediatric ED attendances were recorded (Table 1). The total number of patients with at least one ED attendance was 24,824, distributed among 904 PCPs/GPs across the six LHDs. A total of 71.8% of patients had only one attendance in 2022. Almost 50% of patients attended the ED only once (49.9%), while 3.5% of the total patients (n = 858) accounted for 11.7% of total attendances. Investigation for weekly seasonal effects suggests no important effect of the working day (Figure 1), and this was also confirmed in regression analysis. On the other hand, we found a lower attendance rate in August and September, while peak attendance was observed in October, November, and December (Figure 2).

In Table 2, the number of attendances per groups of variables (District, PCP/GP, individual patient) are distributed and detailed. The total mean of attendances of FU was 1.4, with a maximum of 21 attendances for a single patient. A total of 904 PCPs/GPs had at least one of their patients attending the ED by 2022, with an average of 39.5 children per PCP/GP.

The characteristics of the patients and ED attendances are summarized in Table 3. The regression analysis revealed no significant differences between men and women in terms of FU or non-FU status (*p* = 0.688), working days and festivities (*p* = 0.271), or EMS usage (*p* = 0.363). The FUs seemed to be two years younger than the non-FUs (7.7 and 5.3 years, *p* < 0.001). The most representative age group among non-FUs is middle and early childhood (57.3%), while early childhood and infancy (49.5%) are the most frequent ED users (*p* ≤ 0.001). The attendance parameters are consistent with the demographic distribution of the studied population both for sex and age distribution. Codes 2 and 3 were assigned 19,966 and 7502 (87.1%) times, respectively, among non-FU and 2425 and 1001 (82.4%), respectively, among FUs. The analysis shows a higher classification in codes 1 (highest urgent), 4, and 5 (lower and lowest urgent) of FU patients compared to non-FU patients (*p* ≤ 0.001). The FUs’ attendances are more appropriate than non-FUs’ attendances, at 15.4% vs. 10.2%, respectively (*p* ≤ 0.001).

In Table 4, patients, PCPs/GPs, and districts were used as higher-order nested groups. The appropriateness is associated with male gender (MOR male: 1.06), age < 1 year or age > 15 (MOR < 1 year old: 3.05 and MOR > 15 years old: 3.32), and EMS usage (MOR EMS: 5.65).

Variance analysis shows that the single patient’s characteristics are associated with a greater appropriateness of ED attendance. Meanwhile, PCP/GP or LHD have a weaker influence in determining the appropriateness of pediatric ED attendances (Table 5).

## 4. Discussion

In our study, pediatric FUs are more likely to be two years younger than non-FUs (7.7 and 5.3, *p* = 0.001), while there are no other differences in demographic data analysis regarding access condition. These findings are in line with other Italian studies in which patients who visited the ED for non-urgent attendances are more likely to be younger and male [4,21,27]. Our study shows that up to 62% of pediatric ED attendances were inappropriate for FUs and non-FUs. This result is consistent with other studies results, which reported up to 80% inappropriate attendances for FUs and non-FUs [4,5,28,29]. Among them, non-severe situation are manageable in other settings, meaning that ED is an inappropriate setting of care for such patients. The fact that ED attendances are more likely to occur in winter/spring months suggests that children are directed toward ED in situations that may not actually require high-level care—even more so, since most of such attendances were recorded with low priority codes. This implies that seasonal epidemics (es. flu or pollen reactions), whose clinical impact is usually manageable in primary care setting, still find unmet health needs in EDs. These rates of inappropriate access to EDs are even higher than in adults and indicate a great opportunity to improve the appropriateness of care and patient safety [6]. Finally, in our study, a higher rate of appropriateness for access was observed in FUs compared to non-FUs in Eds; this is likely due to the higher prevalence of serious urgent codes (code 1) compared to non-FU patients.

The nested logistic regression and variance analysis show both PCP/GP and district factors seem to play a small role in determining a greater number of ED attendances. Although the characteristics of a single patient have a greater influence on this phenomenon, probably due to the severity of the disease and the tendency to be admitted to the ward. Young children are at the highest risk of urgent attendance, although it is difficult to distinguish whether this is due to the severity of the disease or to a cautionary approach during triage evaluation [30,31].

Likewise, adult frequent ED users have been profiled as patients who have multiple chronic conditions and mental health issues. Both adult and pediatric FUs have similar characteristics and have also been commonly portrayed as non-urgent utilizers of EDs and for whom more appropriate care should be provided outside the ED setting. However, a large proportion of adult FUs also exhibit multiple visits to the primary care provider [2,8,32]. Barriers to accessing services outside of the ED may differ between pediatric patients and adults. This could depend on the different epidemiological profiles of the children and their health needs, but mostly on the impact of family dynamics and the perceptions of parents and caregivers of EDs as the most appropriate place to receive care. Evidence from the literature has indicated that the caregiver’s poor health literacy, overestimation of clinical urgency, and anxiety are associated with overuse of pediatric ED. In fact, the day of the week is not a significant factor for attendance, and parents of pediatric ED users probably prefer to bypass PCP, even if the most common cause of ED use is respiratory tract infections, which also represent the most common reason for PCP visits [10,33,34]. On the contrary, this may also be related to the difficulties young children experience in communicating their symptoms, determining parents’ concerns about non-urgent conditions such as fever (fever phobia) [35]. This phenomenon may explain the lack of a role for PCP/GP and territorial factors in determining the appropriateness of attendances.

In the experience of Italy and other countries, various intervention strategies have been proposed in the past to reduce the overuse of EDs: the introduction of a fee for non-urgent attendances, a different distribution of PCPs’ offices and their opening times, providing a primary care service in EDs, and the intensification of associated pediatric practice models of work and health promotion with parents. The results and effectiveness of these strategies were generally poor and did not reduce patient compliance [36,37,38,39]. Future perspectives may focus on telemedicine to reduce families’ concerns about their children’s health and better integrate hospital and community care [40,41,42].

### Strengths and Limitations

This is one of the first Italian studies to analyze the rate and characteristics of ED attendances by the pediatric population in a large urban area over the entire year. Furthermore, to our knowledge, for the first time different groups of children have been compared, distinguishing FUs and non-FUs, the appropriateness of ED utilization, and the association with primary care and territorial influence. This study allowed us to define a FU profile of ED for young people. Some limitations should be considered. Disease status and visits to FUs that are children outside the hospital were not investigated in this phase of the study, but they may be addressed in future research. The results on the appropriateness of ED attendances may hardly be generalizable to other countries since Italian regions have different ED organizations, and the definition of appropriateness must be better defined in future studies. Furthermore, due to the limitations of administrative databases, the role of family structure, socioeconomic status, literacy level, distances between the houses of the children and the PCP office, and the PCP model of work are not investigated in this study. Since the study used only administrative and not clinical records, information on specific disease (ICD-9-CM classification, diagnosis, symptoms, etc.) was not accessible to researchers. Lastly, COVID-19 impact was not considered in the study. However, it must be noted that COVID-19, during 2022, was way past its destructive wave impact in Italy. The number of symptomatic cases was low in 2022, and among children, its impact was negligible in terms of direct acute presentation [43].

## 5. Conclusions

Frequent use of EDs is not associated with inappropriate use of EDs among children, mostly due to the characteristics and needs of specific patients. Therefore, it is necessary to adopt strategies and new territorial interventions to allow the appropriate use of health service resources among the general population and to locally provide quality health solutions to children’s health needs. More attention may also be paid to parental attitudes toward general overuse of healthcare resources, particularly EDs. The National Recovery and Resilience Plan (Next Generation EU) on the management of chronic conditions at the territorial level may be an opportunity to help policy makers anticipate and face the needs of specific categories of patients.

## Figures and Tables

**Figure 1 epidemiologia-06-00031-f001:**
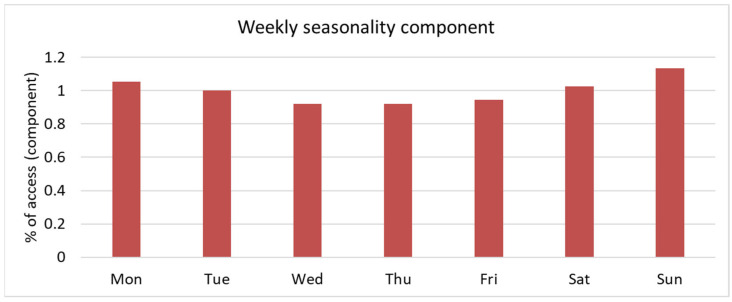
Weekly seasonality component observed for ED attendances.

**Figure 2 epidemiologia-06-00031-f002:**
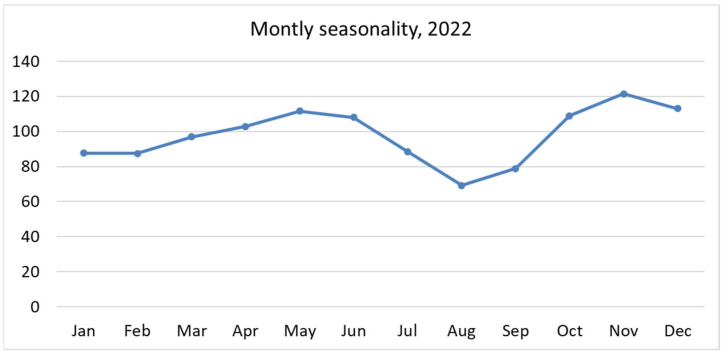
Monthly seasonality component observed for ED attendances.

**Table 1 epidemiologia-06-00031-t001:** Distribution of ED attendance and patients in 2022.

Cumulative Attendance	N° Patients (0–18)	% Patients (0–18)	N° Attendances	% Attendances
1	17,817	71.8%	17,817	49.9%
2	4731	19.1%	9462	26.5%
3	1418	5.7%	4254	11.9%
4	481	1.9%	1924	5.4%
5	218	0.9%	1090	3.1%
6	85	0.3%	510	1.4%
7	29	0.1%	203	0.6%
8	23	0.1%	184	0.5%
9	7	0.1%	63	0.2%
10	4	0.1%	40	0.1%
>10	11	0.1%	144	0.4%
Total	24,824	100%	35,691	100%

**Table 2 epidemiologia-06-00031-t002:** ED attendances grouping distribution in 2022.

Grouping Variable	Groups (N)	Cumulative Patients per Group			Cumulative Attendances per Group		
		Min	Avg	Max	Min	Avg	Max
LHD	6	2995	4137.3	5175	4072	5948.5	7626
PCP/GP	904	1	27.5	261	1	39.5	422
Patient	24,824	-	-	-	1	1.4	21

**Table 3 epidemiologia-06-00031-t003:** Characteristics of ED users and ED attendances in 2022 stratified by user demographic variables and ED attendance parameters.

Variable	Population			
	Total Patients (N = 24,824)	Non-FU (N = 23,966)	FU (N = 858)	*p*-Value
Gender				
Male	14,040 (56.6%)	13,549 (56.5%)	491 (57.2%)	0.688 *
Female	10,784 (43.4%)	10,417 (43.5%)	367 (42.8%)	
Mean age (SD)	7.6 (5.5)	7.7 (5.5)	5.3 (5.5)	<0.001 °
Age group				
Infants (<1)	2491 (10.0%)	2288 (9.5%)	203 (23.7%)	<0.001 *
Toddlers (1)	2017 (8.1%)	1905 (7.9%)	112 (13.1%)	
Early childhood (2–5)	5996 (24.2%)	5775 (24.1%)	221 (25.8%)	
Middle childhood (6–12)	8143 (32.8%)	7966 (33.2%)	177 (20.6%)	
Adolescents (13–15)	3746 (15.1%)	3651 (15.2%)	95 (11.1%)	
Young adults (>15)	2431 (9.8%)	2381 (9.9%)	50 (5.8%)	
	**Attendances**			
	**Total attendance** **(N = 35,691)**	**Non-FU** **(N = 31,533)**	**FU** **(N = 4158)**	
Gender				
Male	15,372 (43.1%)	17,934 (56.9%)	2385 (57.4%)	0.552 *
Female	20,319 (56.9%)	13,559 (43.1%)	1773 (42.6%)	
Age group				
Infancy (<1)	3830 (10.7%)	3078 (9.8%)	752 (18.1%)	<0.001 *
Toddler (1)	3439 (9.6%)	2799 (8.9%)	640 (15.4%)	
Early childhood (2–5)	9114 (25.5%)	7929 (25.1%)	1185 (28.5%)	
Middle childhood (6–12)	10,962 (30.7%)	10,111 (32.1%)	851 (20.5%)	
Adolescent (13–15)	5036 (14.1%)	4591 (14.6%)	445 (10.7%)	
Young adults (>15)	3310 (9.3%)	3025 (9.6%)	285 (6.9%)	
Triage code				
1	2010 (5.6%)	1724 (5.5%)	286 (6.9%)	<0.001 *
2	22,391 (62.7%)	19,966 (63.3%)	2425 (58.3%)	
3	8503 (23.8%)	7502 (23.8%)	1001 (24.1%)	
4	2411 (6.8%)	2058 (6.5%)	353 (8.5%)	
5	376 (1.1%)	283 (0.9%)	93 (2.2%)	
Arrival mode				
EMS	33,797 (94.7%)	29,872 (94.7%)	3925 (94.4%)	0.363 *
NO EMS	1894 (5.3%)	1661 (5.3%)	233 (5.6%)	
Day of the week				
Working day	11,089 (31.1%)	21,705 (68.8%)	2897 (69.7%)	0.271 *
Festivity	24,602 (68–9%)	9828 (31.2%)	1261 (30.3%)	
Appropriateness				
Not appropriate	22,295 (62.5%)	19,823 (62.9%)	2472 (59.5%)	<0.001 *
Appropriate	13,396 (37.5%)	11,710 (37.1%)	1686 (40.6%)	

* χ^2^ test; ° two-sample *t* test.

**Table 4 epidemiologia-06-00031-t004:** Nested logistic regression analysis for the appropriateness of ED attendances and the following variables: gender, age, triage code, and EMS usage.

Variables	Median Odds Ratio	*p*-Value	95% Confidence Interval
**Gender**			
Female	-	-	-
Male	1.06	0.040	1.00–1.13
**Age Group**			
Infancy (<1)	3.05	<0.001	2.73–3.41
Toddler (1)	1.12	0.050	0.99–1.26
Early childhood (2–5)	-	-	-
Middle childhood (6–12)	1.29	<0.001	1.18–1.40
Adolescent (13–15)	2.27	<0.001	2.05–2.51
Young adult (>15)	3.32	<0.001	2.94–3.75
**Day of the Week**			
Workday	-	-	-
Vacation day	0.08	0.007	0.02–0.14
**Arrival Mode**			
Not by EMS	-	-	-
By EMS	5.65	<0.001	4.99–6.38
**Frequent Usage**			
Non-frequent user	-	-	-
Frequent user	1.24	<0.001	1.10–1.39
**Constant Value**	0.28	<0.001	0.21–0.37

**Table 5 epidemiologia-06-00031-t005:** Variance analysis for the appropriateness of ED attendances of LHD, PCP/GP, and single patient.

Variables	Variance	*p*-Value	95% Confidence Interval
District	0.12	-	0.04–0.38
PCP/GP > District	0.03	-	0.02–0.05
Patient > PCP/GP > District	1.41	-	1.24–1.61

## Data Availability

Data supporting the results reported in the article were extracted from Healthcare Emergency Information System. The datasets generated and/or analyzed during the current study are not publicly available due to privacy reason but are available from the corresponding author upon reasonable request.

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
