# Peer review of "The Frequent Use of Emergency Departments Among the Pediatric Population: A Retrospective Analysis in Rome, Italy"

_epidemiologia, 2025, doi:10.3390/epidemiologia6030031_

Round 1
Reviewer 1 Report
Comments and Suggestions for Authors
Thank you for the opportunity to review this manuscript. The manuscript “Frequent use of emergency departments among pediatric population: a retrospective analysis in Rome, Italy” presents a retrospective population study conducted to investigate the characteristics of pediatric attendances in emergency departments in Rome. Also, the study investigates to identify the factors associated with appropriate attendance in the emergency departments. Considering the broader context of presented studies and further impact of the published papers, I have some concerns/suggestions suggested to be addressed before the next steps in the publication.
- The study included 13 emergency departments. How were these departments selected (selection criteria)? How frequent inhabitants visits these facilities in case of emergencies? Also, what about patients visiting private medical care settings? and how are these taken into account?
- The authors have comprehensively analysed the data with respect to study population. However, it would be worthwhile to include (if possible) some data about their parents including occupation, occupational status, education etc.
- Also, it will be important to analyse the data with respect to number of siblings particularly <15 years of age. As frequent visits may be well attributed to household etc. etc. In case the data is unavailable, it would also be ok to include the average household size in the region and number of children below the age of 15.
Minor comments:
- I would suggest replacing “emergency departments” and “Italy” in key words with suitable similar words. As these words have already been used in the title of the manuscript.
Author Response
Major:
The study included 13 emergency departments. How were these departments selected (selection criteria)? How frequent inhabitants visits these facilities in case of emergencies? Also, what about patients visiting private medical care settings? and how are these taken into account?
The study included 13 ED. The reason is described in the Introduction, point 1.2., which describes the setting and population: the totality of ED in the area were included. Please notice that, in Italy, Emergency assistance is publicly provided, free of charge, and in the Lazio region almost 2 million ED accesses per year have been recorded since 2011, with a slight diminution in 2020 ( this is the link to the regional usage monitoring service: https://statistica.regione.lazio.it/statistica/it/lazio-in-numeri/sanita-e-stato-di-salute-della-popolazione/accessi-pronto-soccorso-e-dimissioni) As per Rev 2 suggestion, a time series analysis was added to highlight seasonality effects in ED attendance. We hope the addition to be satisfactory.
The authors have comprehensively analysed the data with respect to study population. However, it would be worthwhile to include (if possible) some data about their parents including occupation, occupational status, education etc.
The study was conducted on anonymous records. This means that we cannot know who the kid actually is, who is parents/relatives are, nor any census data (economic conditions, parents' education level, etc.). However, we acknowledge the question is relevant, so we provided with population-level information in the population description section (point 2.1). We hope the addition to be satisfactory.
Also, it will be important to analyse the data with respect to number of siblings particularly <15 years of age. As frequent visits may be well attributed to household etc. etc. In case the data is unavailable, it would also be ok to include the average household size in the region and number of children below the age of 15.
As per point above.
Minor comments:
I would suggest replacing “emergency departments” and “Italy” in key words with suitable similar words. As these words have already been used in the title of the manuscript.
Thanks for your suggestion. we removed the redundant keywords, and replaced with "frequent user, children, appropriateness, local health authority, emergency medicine, population study, epidemiology"
Reviewer 2 Report
Comments and Suggestions for Authors
Based on my review of the paper, I recommend following changes.
To enhance the quality and depth of the study, it would be valuable to explore any underlying factors contributing to frequent emergency department (ED) visits. For instance, chronic conditions such as asthma, diabetes, or acute bronchitis may play a significant role. Additionally, please consider whether limited access to primary care services contributed to the increased reliance on the ED for non-emergent issues.
It would strengthen the paper to discuss whether patients presenting with less severe conditions could have been appropriately managed in urgent care settings instead of the ED. If applicable, please indicate whether the geographic area where the study was conducted has access to urgent care facilities.
Further, if the data show any seasonal trends—such as an increase in ED visits during winter months due to upper respiratory tract infections—this should be highlighted. If available, information on alcohol or substance use as a potential driver of ED utilization would also provide valuable context.
Finally, please mention whether COVID-19 infection patterns influenced the findings in any way.
Author Response
To enhance the quality and depth of the study, it would be valuable to explore any underlying factors contributing to frequent emergency department (ED) visits. For instance, chronic conditions such as asthma, diabetes, or acute bronchitis may play a significant role. Additionally, please consider whether limited access to primary care services contributed to the increased reliance on the ED for non-emergent issues.
As discussed in limitations, we could not consider the type of illness and the previous health status. This is a consequence of the usage of administrative data instead of clinical records. We considered the possibility that limited access to primary care facilities may impact on ED attendance. We conducted a time series decomposition for weekly seasonality, and found no significant impact of working day on ED usage. This suggest that availability of primary care facilities does not impact on recourse to ED by children.
It would strengthen the paper to discuss whether patients presenting with less severe conditions could have been appropriately managed in urgent care settings instead of the ED. If applicable, please indicate whether the geographic area where the study was conducted has access to urgent care facilities.
We have expanded the discussion highlighting this point. Of course, within the Italian NHS system, patients with less severe conditions may be visited by a PCP and the usage of ED is therefore not appropriate for the used level of care.
Further, if the data show any seasonal trends—such as an increase in ED visits during winter months due to upper respiratory tract infections—this should be highlighted. If available, information on alcohol or substance use as a potential driver of ED utilization would also provide valuable context.
We put an additional analysis considering weekly and monthly seasonal trends, and discussed its inmplications. However, please consider this study had only one year of data available, so any higher-order seasonal component (above the month-level) could not be computed.
Finally, please mention whether COVID-19 infection patterns influenced the findings in any way.
Covid was fairly endemic in Italy during 2022The number of symptomatic cases was low in 2022, and among children, its direct impact was negligible. Cfr. https://www.epicentro.iss.it/coronavirus/bollettino/Bollettino-sorveglianza-integrata-COVID-19_6-dicembre-2022.pdf (Italian documentation regarding COVID-19 epidemic, updated on December 2022) and https://pmc.ncbi.nlm.nih.gov/articles/PMC10540415/.
Round 2
Reviewer 1 Report
Comments and Suggestions for Authors
Thank you for the opportunity to review the revised version of the manuscript entitled “Frequent use of emergency departments among pediatric population: a retrospective analysis in Rome, Italy”. Based on the earlier suggestions and their implementation by the authors, the manuscript has improved manifolds. I have no further comments. I wish best of luck with the further steps in publication.
Reviewer 2 Report
Comments and Suggestions for Authors
Thank you for incorporating the suggestions to enhance the quality of the paper.